# Transcriptome Comparison between Two Strains of *Ustilago esculenta* during the Mating

**DOI:** 10.3390/jof9010032

**Published:** 2022-12-23

**Authors:** Shuqing Wang, Lidan Gao, Yumei Yin, Yafen Zhang, Jintian Tang, Haifeng Cui, Shiyu Li, Zhongjin Zhang, Xiaoping Yu, Zihong Ye, Wenqiang Xia

**Affiliations:** Zhejiang Provincial Key Laboratory of Biometrology and Inspection & Quarantine, College of Life Sciences, China Jiliang University, Hangzhou 310018, China

**Keywords:** *Zizania latifolia*, *Ustilago esculenta*, transcriptome, effector

## Abstract

*Ustilago esculenta* is a smut fungus that obligately infects *Zizania latifolia* and stimulates tissue swelling to form galls. Unlike T-type, MT-type *U. esculenta* can only proliferate within plant tissues and infect the offspring of their host. Production of telispores, haploid life, and plant cuticle penetration are not essential for it, which may lead to the degeneration in these processes. Transcriptome changes during the mating of T- and MT-type *U. esculenta* were studied. The functions of several secreted proteins were further confirmed by knock-out mutants. Our results showed that MT-type *U. esculenta* can receive environmental signals in mating and circumstance sensing as T-type does. However, MT-type *U. esculenta* takes a longer time for conjunction tube formation and cytoplasmic fusion. A large number of genes encoding secreted proteins are enriched in the purple co-expression module. They are significantly up-regulated in the late stage of mating in T-type *U. esculenta*, indicating their relationship with infecting. The knock-out of *g6161* (xylanase) resulted in an attenuated symptom. The knock-out of *g943* or *g4344* (function unidentified) completely blocked the infection at an early stage. This study provides a comprehensive comparison between T- and MT-type during mating and identifies two candidate effectors for further study.

## 1. Introduction

*Ustilago esculenta* is a flavored biotrophic smut fungus that infects *Zizania latifolia* and stimulates tissue swelling to form smut galls on the top internodal region [1,2]. Massive amounts of teliospores are produced in the smut galls at the late stage of infection [3]. In India, *Z. latifolia* is prevalent in swamps and wetlands. Its smut gall with mycelia and teliospores of *U. esculenta* is an edible vegetable locally called “kambong” [4]. For the past 3000 years, uninfected *Z. latifolia* has been cultivated by the Chinese, and its seeds are consumed as a cereal [5,6]. Infected *Z. latifolia* plants without teliospores in galls were found from more than 2000 years ago, and have been further cultivated until now. Instead of smut galls, the cultivated infected plants form white and juicy galls called “Jiaobai” in China [6,7,8].

During the cultivation of infected *Z. latifolia* in China, two kinds of plants were found in the field: those forming white juicy galls and smut galls, respectively. Smut galls are not for eating and are discarded. *U. esculenta* that induces the formation of “Jiaobai” is named MT-type. Only a few teliospores are produced by MT-type at a very late stage of infection. The parasitic mycelium that was produced following germination and mating will induce acute defense responses and cannot infect host plants successfully [9,10]. *U. esculenta* in smut galls with dark sori is named T-type [10]. Genomic comparison showed a great difference between T- and MT-type *U. esculenta*, and an extremely low genetic diversity among different strains of MT-type *U. esculenta* [11]. This implies that MT-type *U. esculenta* is a monophyletic group that is generated by a single incident and evolves alone without gene exchange between T- and MT-types.

T-type *U. esculenta* displays a common life cycle like other smut fungi. It consists of a parasitic dikaryon phase and a saprophytic haploid phase. The mating of two compatible haploid sporidia results in a dikaryotic parasitic mycelium that can penetrate the cuticle of host plants [12]. It infects the host plant at the seedling stage and forms no obvious symptom at this time [13]. At the late stage of infection, *U. esculenta* induces gall formation followed by karyogamy and gives rise to dark teliospores. The teliospores allow fungi to survive in harsh circumstances, germinate under appropriate conditions, and produce haploids after meiosis.

However, for the MT-type, inability to penetrate plant cuticles results in long-term asexual proliferation, in which most parts of the life cycle are missing [9]. Inability to infect the host plant after mating means that teliospores formed by MT-type are not reproducible, and thus useless. MT-type *U. esculenta* in the host plant can only reproduce by its parasitic mycelium. It overwinters in the creeping stem underground and infects the offspring of host plants by extending within plant tissues [14]. Its proliferation strictly depends on the offspring of host plants. The genetic study further showed that *Z. latifolia*, which is infected by the MT-type and cultivated, has a significantly smaller repertoire of immune receptors compared with the wild one [8,9]. The relationship between domestic *Z. latifolia* and MT-type *U. esculenta* is just like mutualism, besides the fact that MT-type *U. esculenta* could be extinct without the cultivation by human beings. MT-type *U. esculenta* provides an excellent example of transformation from a parasite to a mutualistic symbiont.

Biological study showed that both T- and MT-type *U. esculenta* can form teliospores, produce haploids, maintain saprophytic life, and mate. However, the MT-type showed partial deficiency in these processes. The T-type starts to produce sori during the initial stages of gall formation, while the MT-type only produces sori at the late stage [4]. T-type teliospores can germinate within several hours of cultivation, but the germination for MT-type needs more time and obtains a significantly lower germination rate [9]. Haploids of MT-type lost part of the ability for external signal sensing, such as pH and oxidative stress [11]. Finally, haploids of MT-type can mate on the surface of host plants, but no successful infection was observed. Mycelium of MT-type fails to penetrate the cuticle of host plants and is eliminated several days after inoculation. Above all, there is a great difference between T-type and MT-type *U. esculenta* in their biological features. The partial deficiency of the MT-type is mainly because these processes are not necessary for the asexual proliferation of MT-type. During 2000 years of cultivation, a large number of mutations accumulated in genes that are related to these processes and resulted in the degeneration of these genes. Now, it is hard to distinguish which mutation is the first one that triggered the differentiation of the T- and MT-types.

Dimorphism conversion from saprophytic haploid to parasitic dikaryotic mycelium is critical for the colonization of *U. esculenta* and other smut fungi [15,16,17]. It is regulated by mating loci *a* and *b*. Signals from the pheromone and pheromone receptor system are amplified by MAPK and cAMP-PKA signaling pathways and then regulate downstream genes. During this process, more than a quarter of the whole gene set is involved, including genes related to cell cycle, DNA replication, translation, and protein folding [18]. Effectors secreted by *U. esculenta* are also required to further penetrate the plant cell wall and down-regulate the plant defense response. The present study compared the dimorphism conversion process of T- and MT-type *U. esculenta*. Time-resolved transcriptome analysis provides profound insights into the development and pathogenicity of *U. esculenta*. We focused on the genes related to protein folding and modification, carbohydrate metabolism, and secretory effectors.

## 2. Materials and Methods

### 2.1. Strains, Growth Conditions, and Samples Collection

The *Escherichia coli* strain DH5α was used for cloning purposes. Compatible haploids of MT-type *U. esculenta* strains UeMT10 and UeMT46 were isolated from germinated teliospores of white juicy galls. Compatible haploids of T-type strains UeT14 and UeT55 were isolated from germinated teliospores of smut galls. Derived strains from UeT14 and UeT55 are listed in the Appendix A.

Haploids were grown in YEPS liquid medium (yeast extract 10 g/L, peptone 20 g/L, sucrose 20 g/L) at 28 °C on a rotary shaker at 200 rpm. To assess the ability of compatible strains to mate and form dikaryotic mycelium, wild-type strains or mutants were cultured in YEPS liquid medium to an OD_600_ of ~2, harvested by centrifugation for 10 min 2000× *g*, and resuspended in water to an OD_600_ of 1. Compatible haploids were mixed in a 1:1 ratio, and then 2 μL of the mixture was spotted on YEPS solid medium. Plates were incubated at 28 °C for 60 h. The mating process was monitored every 12 h under a microscope. After resuspension and mixture, haploids in water were collected by centrifugation, which is regarded as 0 h post-incubation. Haploids and mycelia on the plate were collected after 24, 36, 48, and 60 h. Samples were frozen immediately in liquid nitrogen and stored at −80 °C until RNA extraction. Three biological repetitions were conducted.

### 2.2. RNA Isolation, mRNA-Seq Library Preparation, and Sequencing

Tissues were ground to a fine powder in liquid nitrogen. Total RNA was extracted using TRIzol reagent (Invitrogen, Carlsbad, CA, USA). RNA concentration was measured using the Qubit RNA Assay Kit on Qubit Fluorometer (Life Technologies, Shanghai, China). RNA degradation was assessed using the RNA Nano 6000 Assay Kit on Bioanalyzer 2100 system (Agilent Technologies, Santa Clara, CA, USA).

An aliquot of 1 μg RNA per sample was used for the library preparation. Sequencing libraries for mRNA were generated using NEBNext Ultra RNA Library Prep Kit for Illumina (New England Biolabs, Beijing, China), following the manufacturer’s recommendations. Briefly, mRNA was first purified based on poly-T magnetic beads and fragmented into ~500bp. cDNA was synthesized using random hexamer primer and converted into blunt ends. Then, NEBNext Adaptor (New England Biolabs, Beijing, China) with hairpin loop structure was ligated, followed by PCR reactions. At last, purified PCR products with adaptors on both sides were ready for sequencing. Sequencing was performed on an Illumina Hiseq platform and more than 50 million 150 bp paired-end reads were generated (Appendix A).

### 2.3. Read Mapping, Normalization, and Statistical Analysis of Differential Gene Expression

Following quality assessment, clean data were obtained by removing reads containing the sequencing adaptor, reads containing poly A or T tails, and low-quality reads. The error rate, Q20, Q30, and GC content of the clean data were calculated. Clean data were then mapped to the reference genome sequence of *U. esculenta* by hisat2 (v2.2.0) with default parameters [11]. HTseq (v0.9.1) was used to count the read numbers mapped to each gene. Fragments per kilobase per million (FPKM) were calculated to estimate gene expression levels [19]. Differentially expressed genes were identified using the DESeq R package (v4.0.0). The resulting *p*-values were corrected using Benjamini and Hochberg’s approach for controlling the false discovery rate. Genes with adjusted *p*-value < 0.05 and abs (log2 (treatment/control)) ≥ 1 were assigned as differentially expressed.

### 2.4. Co-Expression Analysis

The relationships between genes’ expression were evaluated by weighted correlation network analysis (WGCNA) [20]. A total of 6289 genes were analyzed. Log2-transformed FPKM served as input for the network analysis. WGCNA R package (v1.51) was used to create a signed network of a Pearson correlated matrix. To determine a suitable number of clusters, we used the figure of merit analysis and chose to cluster the genes. Average linkage hierarchical clustering of individual modules and gene sets was performed with Cluster (v3.0). The line chart was used to show the trend of gene expression of MT-type and T-type strains.

### 2.5. GO Term Enrichment Analysis

Annotation for Gene Ontology (GO) terms was performed using Blast2GO (v5.2.5). GO enrichment analysis was implemented by the GOseq R package, in which gene length bias was corrected. GO terms with corrected *p*-value < 0.001 were considered for overrepresentation. Visualization of GO term networks was performed with the Cytoscape Enrichment Map plug-in (v3.91) using a Jaccard coefficient cutoff of >0.2. 

### 2.6. Analysis of Transmembrane Domains and Signal Peptides

All proteins encoded by *U. esculenta* were analyzed. The signal peptide of protein was first predicted by SignalP (v5.0). Proteins with signal peptide could be secreted proteins or transmembrane proteins. Their transmembrane helices were then predicted by TMHMM (v2.0). The proteins with signal peptide but not transmembrane helices are regarded as candidate-secreted proteins for further analysis.

### 2.7. Construction of Deletion Strain

A homologous recombination strategy was used for genetic manipulation [21]. For the generation of stable transformants, hygromycin B was chosen as the selection maker. Briefly, the upstream and downstream sequences of the open reading frame were amplified and cloned to a homemade vector, with a hygromycin resistance gene between them. The protoplast of *U. esculenta* was prepared by the treatment of the lywallzyme. For target gene knock-out, the linearized plasmid was transfected into protoplast by PEG-mediated transformation. The candidate transformants were obtained 5–7 days after the incubation on regeneration solid medium (yeast extract 10 g/L, peptone 4 g/L, sucrose 4 g/L, sorbitol 182.2 g/L, agar 15 g/L) with hygromycin B and further selected by PCR. For each gene, two mutated compatible haploids that derive from UeT14 and UeT55 were constructed, respectively. Successful gene deletion was further confirmed by RT-qPCR after the mating of two mutated strains. All primers used for mutant construction are listed in Appendix A.

### 2.8. Plant Inoculation

For inoculation, 20-day-old seedlings of *Z. latifolia* were used. Two compatible strains of wild-type or mutants were cultured in YEPS liquid medium to an OD_600_ of ~2.0. They were harvested by centrifugation for 10 min at 2000 g, resuspended in water to an OD_600_ of 2, and mixed at a 1:1 ratio. Following the description by Zhang et al. [9], about 1 mL of the mixture was injected into the base of a plant without damaging the top region of the stem. Inoculated seedlings were then cultured in a greenhouse under a 12/12 h light/dark cycle at 25 ± 2 °C and 70% relative humidity. Mycelium growth on the leaf surface and in the stem was observed after 5 days, 7 days, or 30 days. Macroscopic symptom of *Z. latifolia* was monitored after 90 days.

### 2.9. Microscopy

The slices were observed by a TCS-SP5 confocal microscope (Leica Microsystems). The nucleus was stained by DAPI. It was excited at 405 nm, and the fluorescence was detected at 440 to 460 nm. The mycelium of *U. esculenta* was stained by WGA-FITC, as described previously [22]. Briefly, plant tissues were fixed in Carnoy’s Fluid overnight, penetrated with 3.75 mM KOH at 85 °C for 60 min, and stained with 10 μg/mL WGA-FITC under vacuum. They were excited at 488 nm, and the fluorescence was detected at 495 to 530 nm. Images were processed using LAS-AF software (Leica Microsystems, Wetzlar, Hesse, Germany, v3.2.0.9652).

### 2.10. Plant Cell Wall Analysis

As described previously, monosaccharide components of the plant cell wall were analyzed by HPLC after hydrolyzation [22]. Plant tissues were ground in liquid nitrogen and dried by lyophilization. An aliquot of 10 mg dry-weight tissue was washed with 1 mL of 70% alcohol three times, and 1 mL 50% chloroform: 50% methanol (*v*:*v*) three times. The starch was further hydrolyzed by amylase; the produced glucose was removed by washing the pellet with water two times. The plant cell wall was then hydrolyzed by TFA; the resulting monosaccharides were analyzed by HPLC.

### 2.11. Statistical Analysis

The symptom of wild-type control and deletion mutants were compared by Chi-square test of independence. All the statistical analyses were performed with SPSS (v20.0).

## 3. Results

### 3.1. Sample Collecting and Transcriptome Sequencing

To analyze transcriptional changes during the dimorphism conversion from saprophytic haploid to parasitic dikaryotic mycelium, the mating process of T- (Figure 1A) and MT-type (Figure 1B) *U. esculenta* were compared on YEPS solid medium. Two compatible haploids of T-type (UeT14 + UeT55) and MT-type (UeMT10 + UeMT46) were mixed and spotted on YEPS solid medium, respectively. In the beginning, only monokaryotic haploids could be found. Conjunction tubes were first observed after 24 h of incubation of T-type haploids (Figure 1A). Dikaryotic cells were formed later at 36 h post-incubation, and mycelia around the colony of T-type could also be observed (Appendix A). However, the mating of MT-type was significantly delayed, in which conjunction tubes were formed 48 h post-incubation, and dikaryotic cells were formed 60 h post-incubation (Figure 1B). The length of mycelium and the percent of dikaryon also showed a partial deficiency of MT-type in mating and mycelia forming (Figure 1C,D).

Haploids and mycelia were collected for transcriptome analysis after 0, 24, 36, 48, and 60 h of incubation. Transcriptome sequencing was performed on the Illumina platform, and more than 50 million 150-bp paired-end reads were generated for each sample. They were mapped to the reference genome of *U. esculenta*. Gene expression levels were first calculated based on counts of exactly mapped reads and then normalized by FPKM (Fragments Per Kilobase per Million). Three biological repeats were conducted. Quality control showed that the error rate of sequencing is lower than 0.03%; the Q20 and Q30 are higher than 97% and 93%, respectively. The expression correlations of the three biological repeats are higher than 0.98 (Appendix A). Differently expressed genes are further analyzed by DESeq.

### 3.2. Gene Clustering and Co-Expression Analysis

The mating of smut fungi will result in dimorphism conversion from saprophytic haploid to parasitic mycelium and start the infection. It is accompanied by dramatic changes in morphology, lifestyle, and nutrient acquisition. To explore the complex transcriptional regulation relationship, we used the expression data of all stages to perform a weighted gene co-expression network analysis (WGCNA) (Appendix A) [20]. If some genes always have similar expression changes in a physiological process, then these genes may be functionally related, and they can be defined as a module. We identified 16 co-expression modules, which were labeled with different colors (Appendix A). The expression profile of each module is depicted in Figure 2. Some modules reflect the difference between T- and MT-type during fungal development. The tan and magenta modules were strongly induced in MT-type at 24 h post-incubation and then down-regulated gradually. Meanwhile, the cyan and grey modules were highly expressed in MT-type at 24 h and 36 h post-incubation and ceased afterward. These four modules keep a low expression level in T-type at all stages. They may be related to the saprophytic life of haploids because MT-type *U. esculenta* still maintains haploids before 36 h post-incubation, while T-type starts mating at 24 h post-incubation. The blue and black modules were induced immediately at 24 h post-incubation in both T- and MT-types. The T-type showed a higher expression level in the blue module, but a lower expression level in the black module. Blue and black modules could be involved in surface sensing or other functions needed in transferring from liquid to the surface of a solid medium. The purple module was up-regulated in T-type when a large scale of parasitic mycelia was formed (60 h post-incubation). This module may correlate with plant cuticle penetration and colonization.

To generate a concise picture of the biological processes associated with pathogenic development, genes in each module were subject to GO enrichment analysis and visualized in a weighted network (Figure 3).

### 3.3. General Changes in Pathogenic Development

In *U. maydis*, the recognition of pheromones secreted by haploids of the compatible mating type will trigger the G2 cell cycle arrest [23,24]. It is sustained until mycelium successfully penetrates the plant cuticle. Genes related to DNA replication, spindle organization, ribosome biogenesis, and primary metabolism are down-regulated when the cell cycle-arrested cell is formed [25]. Then, they display a second expression peak, which is in accord with the release of cell cycle arrest [26]. Our results showed a more complex expression profile in *U. esculenta*. Genes involved in ribosomal assembly and cytoplasmic translation were enriched in the pink and red modules. The pink module is down-regulated in T-type since 24 h post-incubation, when conjunction tubes were formed. However, no obvious down-regulation is observed in MT-type during the mating (48 or 60 h post-incubation). The red module is down-regulated at 24 h post-incubation in T-type, as well as at 60 h post-incubation in MT-type. It corresponds to their mating process. Genes related to spindle organization, cytoskeleton organization, and reproduction are enriched in the turquoise module. They are down-regulated gradually after being transferred to the solid medium. The turquoise module showed a similar expression profile in both T- and MT-type. On the other hand, genes related to vesicle transport are enriched in the black module, which was up-regulated at 24 h post-incubation. It is reported that *U. maydis* can take use of extracellular vesicles for intercellular communication [27].

### 3.4. Protein Folding and Modification

Effectors will be produced in consecutive expression waves after successful mating [28]. The large amounts of effectors impose enormous stress on the endoplasmic reticulum (ER) and protein folding. We found that genes related to protein folding are enriched in the brown module (Figure 2). Their expression level keeps rising during the mating on YEPS solid medium. These genes might be elicited by misfolded proteins in ER through unfolded protein response (UPR). In *U. maydis*, UPR will be activated after the dimorphism conversion, which plays an important role in maintaining ER homeostasis [29]. The balance between UPR and mycelial growth is coordinated by interactions of Cib1 and Clp1 [30,31]. Results showed that protein folding is negatively related to protein translation and mitosis. Genes related to ribosomal assembly and protein translation (red module) are down-regulated immediately after T-type *U. esculenta* start mating (24 h post-incubation), and keep expressing at a low level. Genes related to chromosome organization, spindle organization, and cell division are enriched in the turquoise module. Their expression levels keep reducing since the incubation on YEPS solid medium.

Protein glycosylation is a common posttranslational modification that promotes the correct folding and localization of proteins. It is required for fungal pathogenesis. Most of the modified proteins are membrane proteins or secreted proteins. Genes related to protein modification and glycosylation are enriched in the black module. These genes are up-regulated immediately after 24 h of incubation on the YEPS solid medium and then keep being expressed at a high level. Most of these genes are related to mannosyltransferase and transamidase (Appendix A). The previous study showed that pmt4, a mannose transferase that catalyzes the first step of O-glycosylation, is important for colonization by *U. maydis* [32]. It is responsible for the modification of Msb2, which is a membrane protein that can sense external signals and stimulate the formation of appressorium [33]. As well as O-glycosylation, N-glycosylation has also proved to be necessary for the full virulence of smut fungi [34]. 

Surprisingly, genes related to protein folding and modification showed similar expression profiles in T- and MT-types (Figure 2). The mating in MT-type *U. esculenta* is significantly delayed, which means that MT-type *U. esculenta* has not formed conjunction tubes or started the mating process after 24 or 36 h of incubation. However, these genes were still up-regulated. It seems that these genes are not directly responsible for the mating of compatible haploids. The transition from the liquid medium to the surface of a solid medium may be the direct reason for their higher expression level.

### 3.5. Carbohydrate Metabolic Process

Plant pathogens can secrete hydrolases, polysaccharide lyases, and esterases, which allow the formation, remodeling, or degradation of cell walls [35]. Necrotrophic fungi can deeply degrade plant cell walls for energy supply; however, for biotrophic fungi, it will induce plant defense and should be avoided [36]. We found 36 genes that are related to this process, belonging to different families. Most of these genes are classed into the purple, brown, and yellow modules. No laccase was identified in our analysis. Lack of laccase may make *U. esculenta* poor at degrading lignocelluloses, and only able to infect immature plant tissue.

More than 6 and 17 genes are involved in the hydrolyzation or modification of 1,6-beta-glucan and 1,3-beta-glucan, respectively. 1,6-beta-glucan is an important constituent for the fungus cell wall, but not for the plant cell wall. Genes related to 1,6-beta-glucan were gradually up-regulated during the mating process; this may be responsible for cell wall reconstruction, cell division, mating, and mycelium extension [37]. On the other hand, nearly half of carbohydrate metabolic genes are related to 1,3-beta-glucan. 1,3-beta-glucan is not only utilized by fungi for cell wall composing, but also by host plants to form callose and inhibit fungal infection [38]. Callose accumulation is not observed during the infection of *U. esculenta* (Appendix A). However, to our knowledge, exogenous 1,3-beta-glucanase cannot attenuate callose deposition [39]. The functional redundancy of these genes makes it difficult to study their roles in growth and infection.

*U. esculenta* only recruits a small set of genes for plant cell wall degradation. It includes two endo-1,4-beta-glucanases (*g1206* and *g4902*), a pectin lyase (*g6398*), and an endo-1,4-beta-xylanase (*g6161*). They are up-regulated much more intensively than other carbohydrate metabolic genes at the late stage of mating in T-type, as well as at the late stage of colonization (Figure 4A,B). All of them were clustered into the purple co-expression module. To further analyze plant cell wall degrading during fungal infection, the cell wall of *Z. latifolia* was analyzed by HPLC after being hydrolyzed into monosaccharides. Results showed that the content of xylose is much higher than glucose or other monosaccharides, indicating that xylan is the most important component of the plant cell wall (Figure 4C). The functional study of endoglucanases during the infection process is discussed by Zhang et al. [40]. Knock-out of one endoglucanase did not affect haploid growth but will slow down the infection process and attenuate the final symptom. Interestingly, secreting endoglucanases is still necessary, even though the content of cellulose in plant cell walls is less than 10 percent.

Here, we further studied the role of xylanase (*g6161*) by knocking it out. The deletion mutant of *g6161* showed no obvious difference with the wild type in haploid growth, mating, and mycelium extending on YEPS solid medium (Appendix A). Two combinations of WT (UeT14 + UeT55) and Δ*g6161* (UeT14Δ*g6161* + UeT55Δ*g6161*) were inoculated on seedlings of *Z. latifolia* and observed under the fluorescence confocal microscope. Both WT and Δ*g6161* can form parasitic mycelium, penetrate the cuticle of host plants, and proliferate in plant tissues. However, the biomass of *U. esculenta* in Δ*g6161* is significantly lower than WT at five or seven days post-inoculation (Figure 4D). At the late stage of infection, both combinations can induce the formation of galls with dark teliospores in them, while the symptom of Δg6161 is significantly attenuated (Figure 4E). More than half of the individuals showed no obvious symptoms at the late stage of infection of Δ*g6161* (Figure 4F). White plant tissues without teliospores can be observed in galls formed after Δ*g6161* inoculation; however, galls formed after WT inoculation are full of dark teliospores. These results indicate that xylanase can promote the proliferation of *U. esculenta* inside *Z. latifolia*. To our knowledge, *g6161* is the only xylanase gene expressed during the infection of *U. esculenta*. Surprisingly, the knock-out of *g6161* did not completely prevent the infection. 

### 3.6. Development-Associated Changes of the Secretome

Smut fungi can secrete a large arsenal of several hundred effectors to inactivate plant defense or change the physiology of infected plants. However, most of these effectors are not well understood, due to the lack of conserved domain and the functional redundancy [28]. Here, we first predicted the secretome of *U. esculenta* based on bioinformatic analysis by SignalP and TMHMM. A total of 291 putative secreted proteins were identified (Appendix A). Some of them are homologous to proteins that have been found to act as important virulence effectors. In total, 129 proteins lack any predicted structural or functional domains.

Genes that encode putative secreted proteins disperse in 11 co-expression modules; their expression profile is shown in Figure 5. The most noteworthy feature is that a large set of genes in the purple module keep a low expression level in the beginning and are significantly up-regulated at 48 and 60 h post incubation on YEPS solid medium. A great difference in the expression between T- and MT-type *U. esculenta* is also observed. The purple module contains 222 genes, and 57 of them could encode secreted proteins. This ratio is significantly higher than that of the whole gene set (291 of 7423 genes encode secreted proteins). The purple module includes homologous genes of Axe1 (regulating fungal pathogenicity), xylanase (xylan degradation), and glucanases (cellulose degradation), which are very important for regulating pathogenicity [41,42,43]. It likely represents a specific virulence module.

### 3.7. Putative Effectors

Most effectors of smut fungi are not well understood. To identify effectors that help *U. esculenta* to successfully infect host plants, genes in the purple module are further analyzed. Of 57 genes that encode secreted protein in the purple module, 39 showed no similarity to any functionally annotated gene. In this study, we choose six of them, which are *g943*, *g1072*, *g4195*, *g4344*, *g4740*, and *g5676*. According to transcriptome analysis, they are most abundantly expressed at 60 h post-incubation in T-type *U. esculenta*. Their expression profile is further studied by RT-qPCR after being inoculated to *Z. latifolia*. They are also significantly up-regulated during the infection process (Appendix A). Homologs of these genes can be found in several smut fungi including *U. maydis*, *U. hordei*, *Sporisorium reilianum*, etc., but not in species other than smut fungus (Appendix A). 

These genes are knocked out by homologous recombination, respectively. All mutants are derivates of two compatible haploids of T-type *U. esculenta* (UeT14 and UeT55). Mutants of MT-type are not studied, because MT-type cannot successfully infect host plants. Derived strains are listed in Appendix A. The deletion mutants of *g943* and *g4344* showed no obvious difference with the wild type in haploid growth, mating, and mycelium growth on YEPS solid medium (Appendix A and Figure 6A). The deletion of *g1072*, *g4195*, *g4740*, or *g5676* did not affect the saprophytic growth of *U. esculenta*, but resulted in a mating deficiency. The mutants of Δ*g1072*, Δ*g4195*, Δ*g4740*, or Δ*g5676* cannot form parasitic mycelium on YEPS solid medium.

The deletion mutants of these genes are then inoculated to seedlings of *Z. latifolia*, as well as T14 and T55 strains which are recruited as the positive control. Parasitic mycelia can be formed by wild type and mutants of *g943* and *g4344*, but not by mutants of *g1072*, *g4195*, *g4740*, or *g5676*. This is in accord with their mating performance on the YEPS medium. Wild-type *U. esculenta* can colonize the top region of the plant stem and proliferate in it 30 days after the inoculation. However, no mycelia can be found in plant stem for all the mutants. Parasitic mycelia of mutants on plant surfaces are also scavenged, probably due to plant defense response or the lack of nutrition (Figure 6B). These results indicate that all mutants of six genes fail to infect host plants. The final symptom of *Z. latifolia* further confirmed this (Figure 6C, Appendix A). More than 80 percent of plants form smut galls after being inoculated with wild-type strains, but no smut gall was formed after being inoculated with mutants (Figure 6D).

Above all, these results showed that *g943*, *g1072*, *g4195*, *g4344*, *g4740*, and *g5676* are essential for *U. esculenta* to complete its life cycle. They function in different stages. Deletions of *g1072*, *g4195*, *g4740*, or *g5676* lead to mating deficiency, subsequently making mutants fail to infect host plants. Interestingly, they still keep a high expression level after successful mating (60 h post incubation on YEPS medium). They may also play important roles in the infection process like umFly1 (funglysin1), a dual-function metalloprotease that is required in both mycelial growth and plant defense repression [43]. However, this is complicated to prove. On the other hand, both *g943* and *g4344* are only required in the parasitic stage. They may act as effectors to interact with host plants and promote the infection process.

## 4. Discussion

The long-term asexual proliferation of *U. esculenta* in the host plant results in a special mutualistic symbiont named MT-type. Teliospore formation, mating, or plant cuticle penetration are not required by MT-type *U. esculenta*. MT-type *U. esculenta* makes use of dikaryotic mycelium to overwinter in plant tissues and reach the offspring of its host plants [9]. It leads to the accumulation of mutations in genes related to these processes and the degeneration of MT-type *U. esculenta.* We compared the transcriptome difference between T- and MT-type *U. esculenta* during the mating.

The cell cycle of *U. maydis* will be arrested at the G2 phase since the pheromone recognition [24]. Cytoplasmic fusion of smut fungi will not be followed by karyogamy, resulting in a dikaryotic parasitic mycelium. The *b* mating-type locus and a series of factors including Rbf1, Clp1, Cib1, Biz1, and Cdc25 play important roles in this process [22,44]. Here, we showed that genes related to spindle organization, cytoskeleton organization, and reproduction were down-regulated similarly in both T- and MT-type *U. esculenta* (turquoise module), even though they have different timetables for mating. The G2 cell cycle arrest is a prerequisite for cytoplasmic fusion. It suggested that haploids of MT-type might receive the signals from the pheromone pathway and trigger G2 cell cycle arrest as T-type haploids do. However, MT-type *U. esculenta* takes a longer time for conjunction tube formation and cytoplasmic fusion. Some genes were only up-regulated in MT-type before conjunction tubes were formed (tan, magenta, cyan, and grey modules). Genes related to homologous recombination were enriched in the cyan module. No enrichment in any GO or KEGG term was found for tan, magenta, and grey modules. It is not certain whether these genes are needed in the saprophytic life of haploids or the mating process. It can also be explained by the degeneration or dysregulation of MT-type *U. esculenta* during mating.

Genes involved in protein folding and modification also showed a similar expression pattern in T- and MT-type *U. esculenta* (brown module). Unlike G2 cell cycle arrest, these genes are usually activated by misfolded proteins, such as consecutive waves of effectors secreted by parasitic mycelia of *U. maydis* [28,30,31]. During the colonization of *U. maydis*, genes related to protein folding are co-expressed with many genes encoding secreted proteins [25]. However, we did not observe any correlation between genes that help protein folding (brown module) and genes encoding secreted proteins (purple module) in *U. esculenta*. The up-regulation of genes involved in protein folding and modification may not be induced by secreting a large number of proteins. We supposed that these genes might be induced by the transition from the liquid medium to the surface of a solid medium. The modification of membrane proteins, such as msb2, plays an important role in surface sensing [45]. We found that the haploid of T- and MT-type *U. esculenta* can form a mycelium-like structure on the plant surface, but cannot successfully infect host plants (Appendix A). This suggested that MT-type *U. esculenta* maintains the ability for surface sensing. Environmental signal sensing might still play an important role in the parasitic life of MT-type *U. esculenta* in plant tissues, even though the circumstance is much more stable than outside.

Finally, we studied the secretome of *U. esculenta*. Smut fungi can secrete various proteins to degrade the plant cell wall, suppress plant defense response, or hijack the metabolism of plant cells [25]. The function of one xylanase (*g6161*) and six undefined genes (*g943*, *g1072*, *g4195*, *g4344*, *g4740*, and *g5676*) were studied by knocking out, respectively. Xylan is the major component of the cell wall of *Z. latifolia*, and *g6161* is the only expressed xylanase in *U. esculenta* during colonization. However, the knock-out of *g6161* did not completely block the infection. On the other hand, glucan is a minor component of the cell wall of *Z. latifolia.* At least two beta-1,4-glucanase have been identified in *U. esculenta* and knock-out of one of them will attenuate symptom development [38,46]. This could be explained by the cross-link between cellulose and hemicellulose in the plant cell wall [47]. The functional redundancy of xylanase and glucanase is fascinating. We also identified two candidate effectors that are essential in the stage of parasitic mycelium infection.

In summary, we studied the transcriptome change of T- and MT-type *U. esculenta* during mating. Though the mating of MT-type *U. esculenta* did not take place for up to 2000 years, compatible haploids can mate to form dikaryotic mycelium with partial deficiency. It can receive environmental signals and induce G2 cycle arrest, but conjunction tube formation and cytoplasmic fusion are significantly delayed. We also identified several important genes in the infection process which are worthy of further study.

## Figures and Tables

**Figure 1 jof-09-00032-f001:**
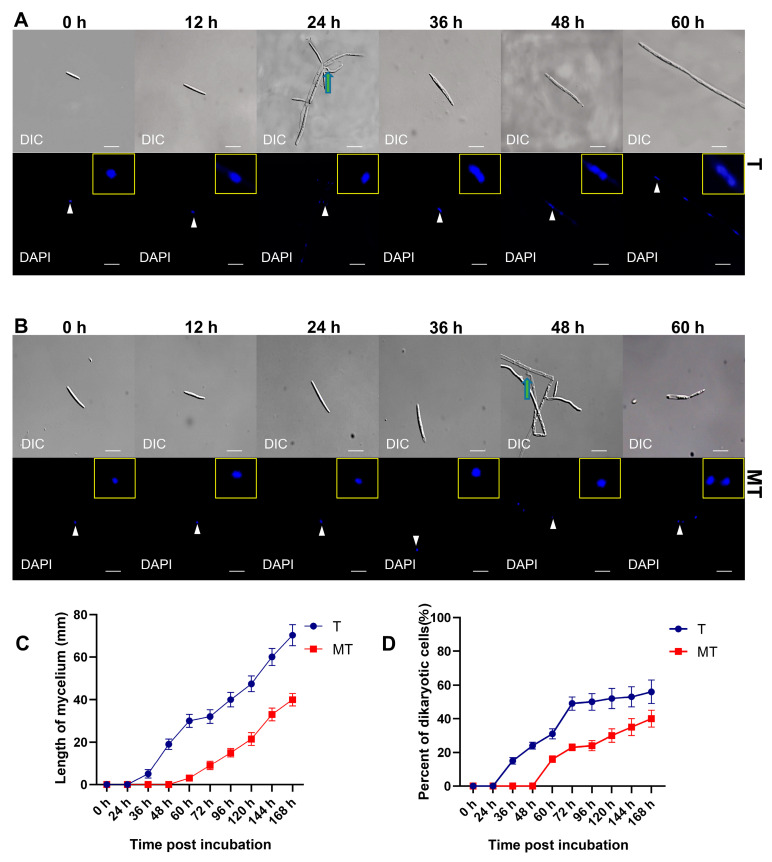
Comparison of the mating process of T- and MT-type *U. esculenta*. (**A**) The mating process of T-type *U. esculenta* on YEPS solid medium. (**B**) The mating process of MT-type *U. esculenta* on YEPS solid medium. Differential interference contrast (DIC) was used to observe the structure of mycelia. Nuclei were stained by DAPI. The conjunction tube is marked with a green arrow. The nucleus, which is enlarged in the upper right of the picture, is marked with a white arrow. The scale bar represents 70 μm. (**C**) Mycelia length around colonies of T- and MT-type *U. esculenta* were monitored after 0 to 168 h incubation on YEPS solid medium (Mean ± SD, number of replications = 8). (**D**) Percent of dikaryotic cells. Nuclei were stained by DAPI to distinguish dikaryotic or monokaryonic cells (Mean ± SD, number of replications > 200).

**Figure 2 jof-09-00032-f002:**
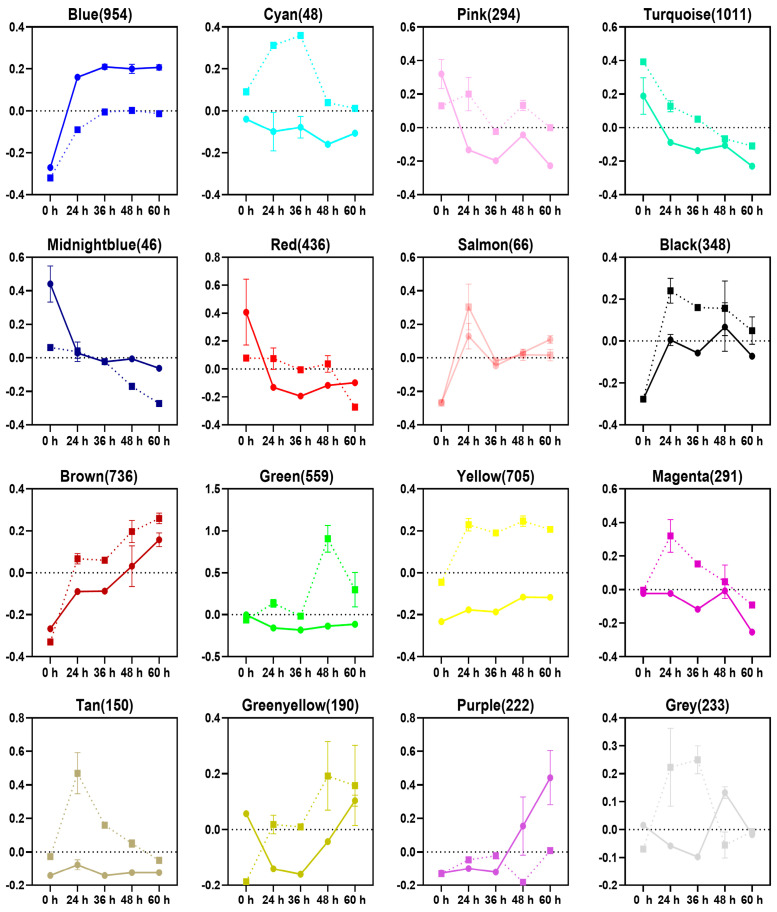
Expression profile of co-expression modules during the mating of *U. esculenta*. The RNA-seq expression data set was subjected to WGCNA to define co-expressed modules. Squares and dotted lines represent MT-type *U. esculenta*, and circles and solid lines represent T-type *U. esculenta*. The vertical axes indicate log_2_ expression levels relative to the mean expression across all stages. Error bars indicate the standard deviation of three biological replicates. The horizontal axes indicate the stages, i.e., 0, 24, 36, 48, and 60 h post-incubation.

**Figure 3 jof-09-00032-f003:**
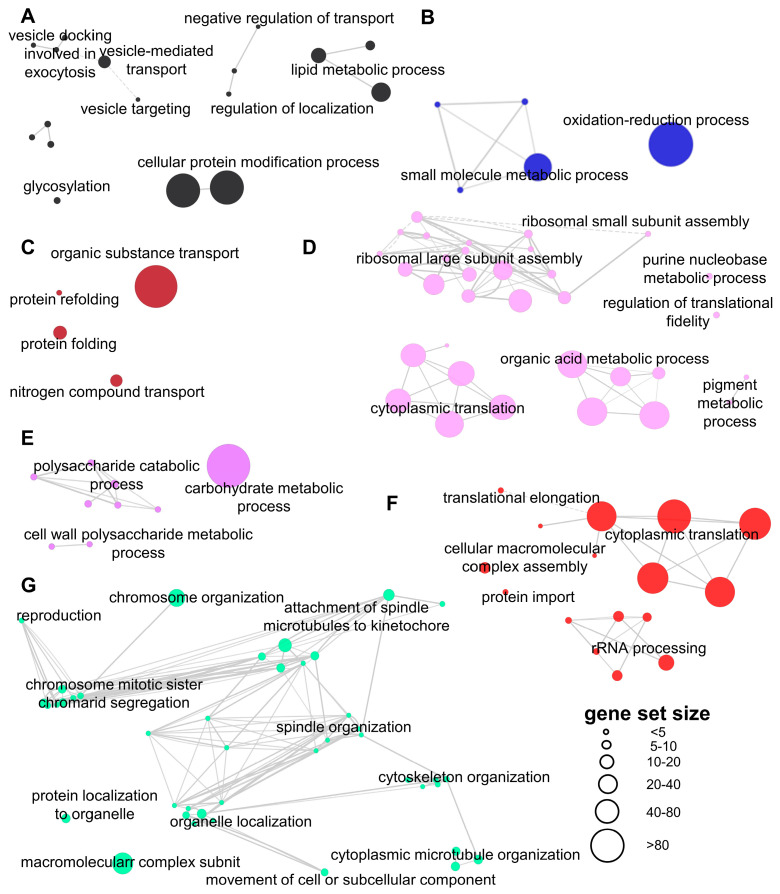
Biological Processes (BP) enriched in selected co-expression modules. GO enrichments in BP were analyzed for the black (**A**), blue (**B**), brown (**C**), pink (**D**), purple (**E**), red (**F**), and turquoise (**G**) modules. Corrected hypergeometric *p*-values < 0.001 were considered for overrepresentation. Each significantly enriched gene set is represented by a node. Node size is related to the number of genes in the gene set.

**Figure 4 jof-09-00032-f004:**
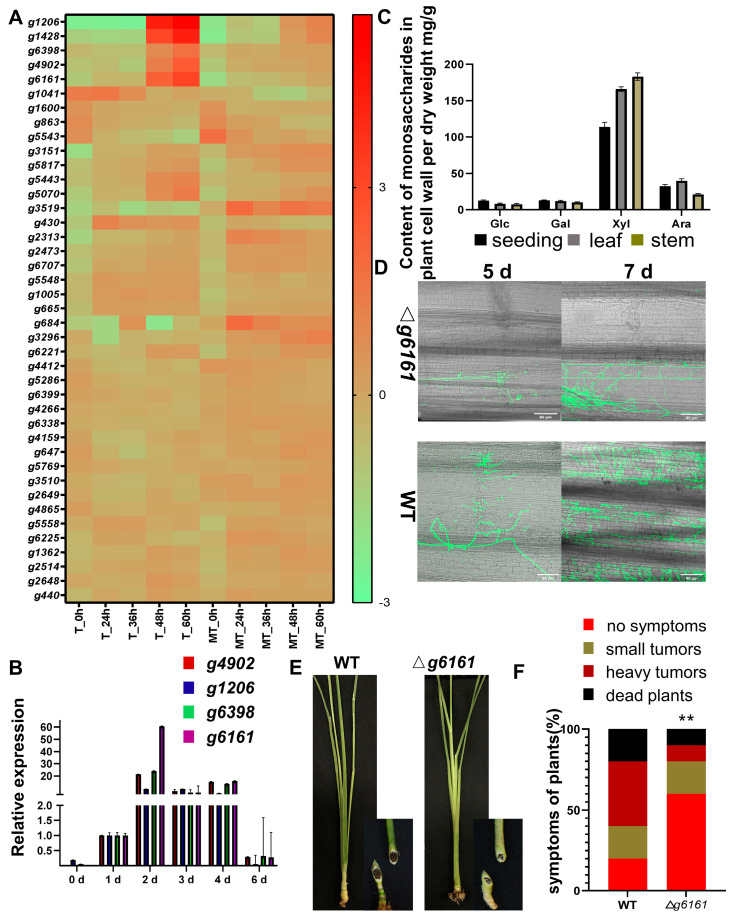
Expression pattern and virulence function of genes related to carbohydrate metabolism. (**A**) The heat map shows the expression profiles of the carbohydrate metabolism-related genes in *U. esculenta*. Log2 expression values are visualized relative to the mean expression level across all stages. (**B**) The expression trend of genes encoding endo-1,4-beta-glucanases (*g1206* and *g4902*), pectin lyase (*g6398*), and endo-1,4-beta-xylanase (*g6161*). Two compatible haploids of T-type (UeT14 + UeT55) were mixed and inoculated to host plants. Gene expressions were evaluated at 0, 1, 2, 3, 4, and 6 days post-inoculation by RT-qPCR (Mean ± SD, number of replications = 8). (**C**) Monosaccharide content in plant cell wall per dry weight of different tissue (seedings, leaves, stems) of *Z. latifolia* (Mean ± SD, number of replications = 6). The plant cell wall was hydrolyzed by trifluoroacetic acid and analyzed by HPLC. (**D**) Observation of infection status on the leaf surface. The mixtures of compatible strains were injected into the stems of *Z. latifolia*. Samples were collected at 5 and 7 days post-inoculation, stained with WGA-FITC, and analyzed by laser scanning confocal microscopy. The combination of Δ*g6161* (*UeT14*Δ*g6161* and UeT55Δ*g6161*) was studied, and wild-type strains (UeT14 and UeT55) were set as a positive control. Green fluorescence indicates mycelia of *U. esculenta.* The scale bar represents 90 µm. (**E**) The symptom of representative plants at 90 days post-inoculation. The cross-section of the stem is shown in the lower right corner. (**F**) The symptoms of inoculated host plants were scored 90 days later according to severity (number of replications = 100). Statistically significant difference with wild-type control: ** *p* < 0.01.

**Figure 5 jof-09-00032-f005:**
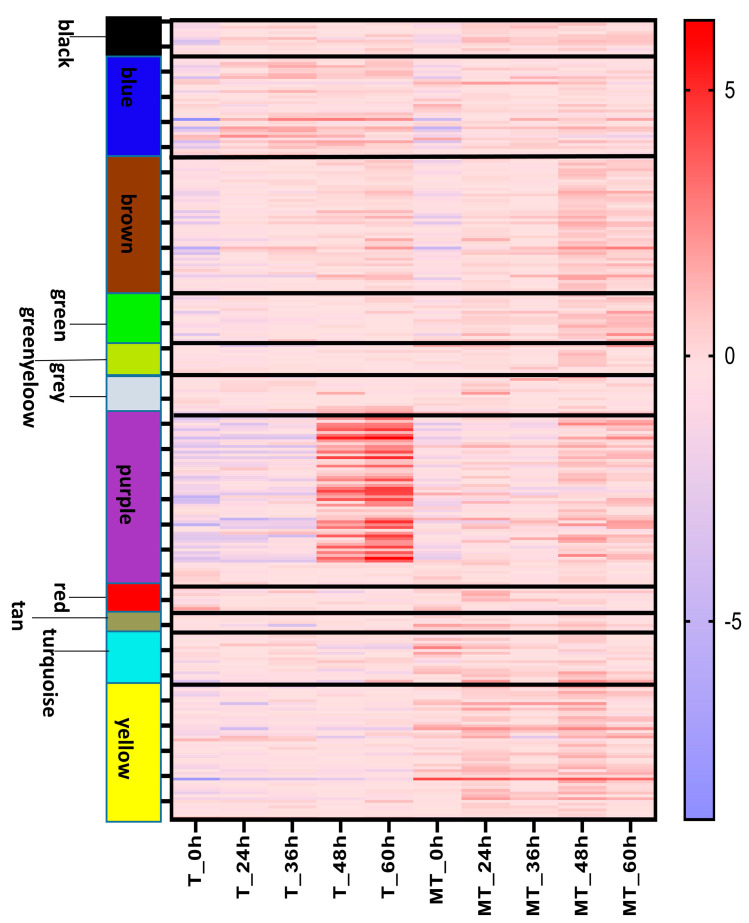
Expression of the *U. esculenta* secretome. For each co-expression module, the heat map shows the expression of genes encoding putative secreted proteins. Log2 expression levels are visualized relative to the mean expression across all stages of T- and MT-type *U. esculenta*.

**Figure 6 jof-09-00032-f006:**
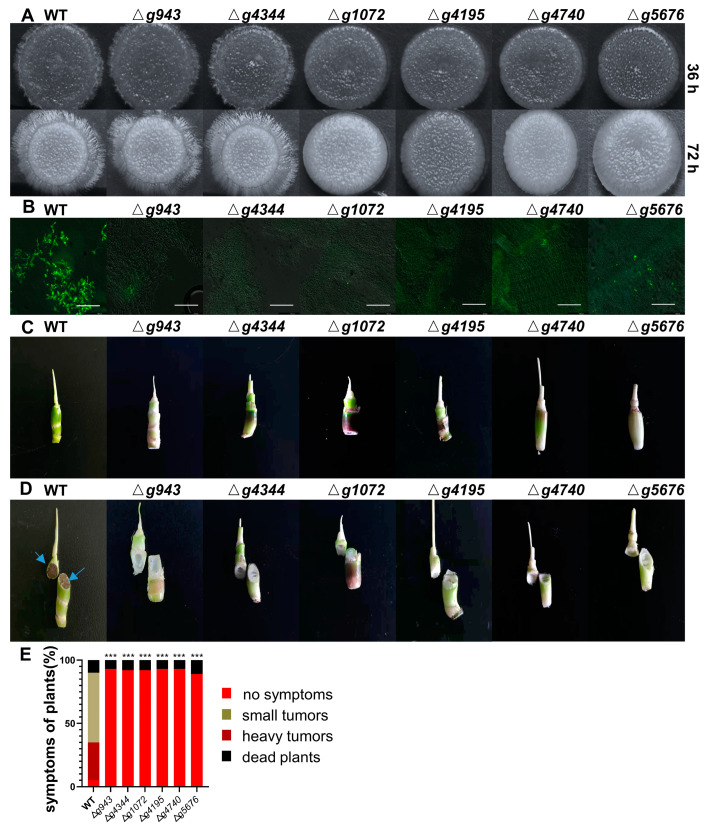
Role of secreted proteins in virulence. Deletion mutants of *g943*, *g4344*, *g1072*, *g4195*, *g4740*, and *g5676* were studied, and wild-type (UeT14 and UeT55) was set as positive control. (**A**) Typical colony mating morphology of deletion mutants under a stereomicroscope at 36 h and 72 h after culture. (**B**) Observation of infection status in the stem. The mixtures of compatible strains were injected into the stems of *Z. latifolia*. Samples were collected at 30 days post-inoculation, stained with WGA-FITC, and analyzed by laser scanning confocal microscopy. Green fluorescence indicates mycelia of *U. esculenta*. The scale bar represents 75 µm. (**C**) The symptom of representative plants at 90 days post-inoculation. (**D**) The cross-section of the stem of representative plants at 90 days post-inoculation (The blue arrows refer to teliospore produced by *U. esculenta*). (**E**) The symptoms of inoculated host plants were scored 90 days later according to severity (number of replications = 100). Statistically significant difference with wild-type control: *** *p* < 0.001.

## Data Availability

The data supporting this study’s findings are available from the corresponding author, Wenqiang Xia, upon reasonable request.

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
