# Peer review of "Transcriptome Comparison between Two Strains of Ustilago esculenta during the Mating"

_jof, 2022, doi:10.3390/jof9010032_

Round 1
Reviewer 1 Report
In the manuscript submitted by Wang et al., the authors compared gene expression patterns during mating process of Ustilago esculenta strains. One strain (termed T) is virulent, but the other strain (termed MT) is nearly avirulent possibly due to endophytic circumstances for long period. Putative secreted proteins are also detected and it is likely that they have virulence functions.
In my view, the manuscript is very nicely written and presented. I consider that it warrants publication. However, I recommend the authors to mention the following minor points in the revised manuscript.
I recommend the authors to assess virulence of deletion mutants with proper statistics.
I would like to ask the authors to comment on any homologs of 6 secreted proteins in other smut fungi (i.e. are these conserved among smut fungi ? or unique to U. esculenta ?)
In terms of the use of “effector”, can you really call the proteins as effector, of which deletion mutant shows reduced mating efficiency ?
Reviewer 2 Report
The authors presented a very elegant study by which they comparatively analyzed the transcriptome of two strains of Ustilago esculenta during the mating. Ustilago esculenta is a smut fungus that infects Zizania latifolia as a obligate parasite and causes galls. Their study compared the dimorphism conversion process of T- and MT-type U. esculenta. They conducted time-resolved transcriptome analysis providing insights into the development and pathogenicity of U. esculenta. They also focused on identifying the genes related to protein folding and modification, carbohydrate metabolism, and secretory effectors. The experimental design for hypothesis testing was well structured. The whole presentation, but specially the results section and the illustrations of the processess studied are simply impecable. As major observations, the authors detected, using a the purple co-expression module, a large number of genes encoding secreted proteins. These genes were significantly up-regulated in the late stage of mating in T-type U. esculenta, indicating their relationship with the host plants during the infection process. The knock-out of genes such as g6161 (xylanase) resulted in an attenuated symptom. The knock-out of g943 or g4344 (with function yet unidentified) completely blocked the infection at an early stage. In summary, their study provided a comprehensive comparison between T- and MT-type during mating and identified two candidate effectors. My recommendation is for acceptance as presented.
